# Parent-Reported Child and Parent Quality of Life during COVID-19 Testing at an Australian Paediatric Hospital Outpatient Clinic: A Cross-Sectional Study

**DOI:** 10.3390/healthcare11182555

**Published:** 2023-09-15

**Authors:** Natasha K. Brusco, Margie Danchin, Jennifer J. Watts, Carol Jos, Myles Loughnan, Tria Williams, Julie Ratcliffe, Monsurul Hoq, Shidan Tosif, Jessica Kaufman

**Affiliations:** 1Rehabilitation, Ageing and Independent Living (RAIL) Research Centre, School of Primary and Allied Health Care, Monash University, Frankston 3199, Australia; 2Vaccine Uptake Group, Murdoch Children’s Research Institute, Melbourne 3052, Australia; margie.danchin@rch.org.au (M.D.); myles.loughnan@gmail.com (M.L.); tria.williams@mcri.edu.au (T.W.); jess.kaufman@mcri.edu.au (J.K.); 3Department of General Medicine, The Royal Children’s Hospital, Melbourne 3052, Australia; shidan.tosif@rch.org.au; 4Department of Paediatrics, School of Population and Global Health, The University of Melbourne, Melbourne 3052, Australia; 5School of Health and Social Development, Deakin University, Burwood 3125, Australia; j.watts@deakin.edu.au; 6Infection and Immunity, Murdoch Children’s Research Institute, Melbourne 3052, Australia; 7Health and Social Care Economics Group, Caring Futures Institute, Flinders University, Adelaide 5001, Australia; julie.ratcliffe@flinders.edu.au; 8Clinical Epidemiology and Biostatistics Unit, Murdoch Children’s Research Institute, Melbourne 3052, Australia; monsurul.hoq@mcri.edu.au; 9RCH National Child Health Poll, The Royal Children’s Hospital, Melbourne 3052, Australia; 10Department of Paediatrics, The University of Melbourne, Melbourne 3052, Australia

**Keywords:** quality of life, COVID-19, respiratory infection, child, cross-sectional study

## Abstract

Globally, we have seen a drop in adult and child quality of life (QOL) during the COVID-19 pandemic. However, little is known about adult or child QOL during the height of the pandemic in Australia and the impact of government-imposed restrictions, specifically attending school on-site versus home schooling. Our study aimed to establish if QOL in children and parents presenting to a Respiratory Infection Clinic in Victoria, Australia, for COVID-19 PCR testing differed from pre-pandemic population norms. We also explored whether on-site versus home schooling further impacted QOL. Following the child’s test and prior to receiving results, consenting parents of children aged 6 to 17 years old completed the Child Health Utility 9 Dimension (CHU9D) instrument on their child’s behalf. Parents of children aged birth to five years completed the EuroQOL 5-Dimension 5-Level (EQ-5D-5L) instrument on their own behalf (cross-sectional study). Data analyses utilised quantile regression, adjusting for the child’s age, COVID-19 symptoms, gender and chronic health conditions. From July 2020 to November 2021, 2025 parents completed the CHU9D; the mean age for children was 8.41 years (±3.63 SD), and 48.4 per cent were female (*n* = 980/2025). In the same time period, 5751 parents completed the EQ-5D-5L; the mean age for children was 2.78 years (±1.74 SD), and 52.2 per cent were female (*n* = 3002/5751). Results showed that QOL scores were lower than pre-pandemic norms for 68 per cent of the CHU9D group and 60 per cent of the EQ-5D-5L group. Comparing periods of on-site to home schooling, there was no difference between the median QOL scores for both CHU9D (0.017, 95% CI −0.05 to 0.01) and EQ-5D-5L (0.000, 95% CI −0.002 to 0.002). Our large-scale study found that while QOL was reduced for children and parents at the point of COVID-19 testing during the pandemic, differing levels of government-imposed restrictions did not further impact QOL. These unique insights will inform decision-making in relation to COVID-19 and future pandemics.

## 1. Introduction

Early responses to the COVID-19 pandemic in most countries included lockdown measures to reduce community transmission of the virus. In the Australian state of Victoria, people experienced some of the toughest lockdowns globally. These included stay-at-home measures, school closures and restrictions regarding travel [1]. Victoria experienced multiple ‘waves’ of infection throughout 2020 and 2021, and restrictions changed frequently over this period [1]. As the pandemic progressed, concerns increased regarding the impact of COVID-19 testing and extended school closures—with subsequent periods of home learning—on both parent and child health-related quality of life (QOL) [2].

Globally, we have seen a significant drop in adult and child QOL utility scores during the pandemic era [3,4,5,6,7,8]. A range of factors have been found to increase the negative impact the COVID-19 pandemic has on QOL. Social factors include job loss, lower income, location, nationality, medical co-morbidities such as anxiety disorders and chronic disease states, as well as COVID-19-specific factors such as perceived risk of COVID-19 infection, presence of symptoms, contact with infected persons and a lack of access to information regarding the pandemic [6,7,9,10,11,12]. Almost universally, individuals living under stay-at-home orders have suffered a loss of QOL since the beginning of the pandemic [13,14]. Although there is no data on the impact of undergoing polymerase chain reaction (PRC) testing on QOL, there is potential concern about possible infection and anticipation while awaiting the result.

While children and families in Victoria have been significantly affected by some of the most stringent COVID-19-related restrictions and long periods of remote school learning globally, little is known about child and parent QOL during the height of the pandemic in the Australian context, nor about the impact of home schooling during the stay-at-home orders [15]. It is important to investigate the impact on children’s quality of life because the health state of a child can impact the QOL of the parents [16]. It is also important to investigate the effect of home schooling on QOL as this was a core strategy for the Victorian stay-at-home orders and placed much stress on the family unit during the COVID-19 pandemic [15,17].

The aim of this study was to determine parent-reported child quality of life and parent quality of life during COVID-19 public health restrictions. Measurements were taken at the point of child and parent COVID-19 testing at an Australian paediatric hospital outpatient clinic. The study objectives were to (i) report parent-reported QOL in children aged 6 to 17 years; (ii) report QOL in parents of children aged birth to 5 years; (iii) investigate the effect of schooling location on QOL (on-site or home-based) as impacted by government-imposed COVID-19 pandemic restrictions; and (iv) compare QOL results to pre-pandemic population norms. It was hypothesised that home-based schooling, compared to on-site schooling, would have a negative impact on QOL and that QOL during the pandemic would be generally less than pre-pandemic population norms.

## 2. Methods

### 2.1. Study Setting

We completed this study at The Royal Children’s Hospital Respiratory Infection Clinic (RIC) in Melbourne, Victoria. The RIC was established in May 2020 and was a large testing site in Melbourne for children, particularly those under the age of five years who required testing according to state guidelines [18]. In Victoria, government-imposed restrictions relating to stay-at-home orders, including home schooling, were the most stringent in the country. The stay-at-home orders were ceased and reenacted multiple times throughout the COVID-19 pandemic [15]. Unique factors for this study are as follows:(1)This single-centre study was conducted at Victoria’s largest tertiary paediatric hospital, and this hospital was the only dedicated testing centre for children under five years of age in Victoria. As such, it captured the largest cohort of children under 5 years old in urban Melbourne and was also accessed by many regional families due to the anticipated distress of testing in young children;(2)We acknowledge that this study is not representative of children and families across Australia, and this was not the intention. Each state and territory in Australia experienced differing levels of COVID-19 waves of infection and subsequent public health restrictions. These results are specific to children and families in metropolitan Melbourne, Victoria, who visited the paediatric hospital outpatient clinic for COVID-19 testing and experienced the most stringent and long-lasting lockdowns in Australia. The experience of children and families would almost certainly have been different in the other Australian states, especially Western Australia, where no lockdowns were applied. These data are representative of populations undergoing harsh pandemic restrictions and the potential impact on QOL and wellbeing;(3)Our data have been time-stamped to allow comparison between periods of home and on-site schooling, which has generated intense debate in the media and between political parties in Australia and globally and, as such, will be of broad interest.

### 2.2. Study Design

This is a cross-sectional cohort study based on a single brief survey instrument to assess parent or child QOL and was administered soon after PCR COVID-19 testing. A sample of those who completed the QOL assessment were later invited to enrol in the COVID Wellbeing Study, a mixed-methods longitudinal cohort study investigating the immediate and longer-term health and wellbeing impacts of COVID-19 on children and families tested for COVID-19 at the RIC. The findings of the COVID Wellbeing Study will be published separately.

### 2.3. Participants and Consent

The sample included children and parents attending the RIC for respiratory SARS-CoV-2 PCR testing between July 2020 and November 2021. Parents completed the QOL instrument on behalf of themselves if the child was under five years old, or as a proxy for their child if the child was six years or older. Parents may or may not have been tested at the same time. All parents attending the RIC were asked, upon registration, for their consent to be contacted about COVID-19 research. Those who consented were sent a link via email to complete an online QOL assessment immediately following COVID-19 testing and prior to receiving test results. Consent to participate was implied by survey completion. The survey was only available in English.

If parents presented to the RIC Clinic for multiple episodes of their child’s COVID-19 testing during the data collection period, they were invited to complete the survey each time. The analysis has considered each response independent of the previous response, as it is not known how many repeat visits occurred due to the nature of the data or if it was the same parent who completed the survey on subsequent visits. If a parent did not complete the survey, they received three follow-up reminders.

Sample size: The sample size for this cross-sectional cohort study is all children and parents who attended the COVID-19 outpatient testing site at the state’s largest paediatric hospital between July 2020 and November 2021 and provided consent for study participation. It was not possible to establish an a priori sample size for this study, as this study commenced at the beginning of the pandemic in Australia when we were unable to predict how many children and parents would attend the testing site as the prevalence of the disease was unknown.

Sample selection: Reference population were people living in the City of Greater Melbourne in 2020/21 = 816,536 families with an average of 1.8 children aged 15 or less (https://www.abs.gov.au/census/find-census-data/quickstats/2021/2GMEL; accessed on 1 June 2022). Study population were people who attended the COVID-19 outpatient testing site between July 2020 and November 2021 (*n* = 70,000). Study participants were those from the study population who consented to participate in the survey (Figure 1).

As all parents who attended the testing site with their child(ren) and provided consent to be sent a link via email to consider participating in the research were sent the study link, there was no sample-selection bias (as all parents were approached, and all parents who provided consent to send a link via email were invited). However, there is the possibility of self-selection bias with respect to the individuals who completed the survey. It is also noted that the identification of home vs. on-site schooling was done based on the data date stamp at the data analysis stage and was unlikely to be influenced by self-selection.

### 2.4. Data Collection

Data were collected over 18 months, between July 2020 and November 2021, representing the early to mid-stages of the COVID-19 pandemic. The surveys were administered using REDCap [19,20], a secure web-based platform to capture data for research, hosted at Murdoch Children’s Research Institute, Melbourne. Parents of children aged birth to five years old completed the EuroQOL 5-Dimension 5-Level (EQ-5D-5L) [21] questionnaire on their own behalf. Parents of children aged 6 to 17 years old completed the validated proxy-reported Child Health Utility 9 Dimension (CHU9D) [14] questionnaire on behalf of their child. Most questionnaires were completed at the RIC during the visit, as compared to being completed at home after the RIC visit.

### 2.5. Outcome Measures

The EQ-5D-5L instrument is the world’s most widely applied preference-based measure of health-related QOL and includes five key dimensions: mobility, self-care, usual activities, pain/discomfort and anxiety/depression. Participants are asked to rate their level of health across five levels within each of these domains (from ‘no problems’ to ‘extreme problems’), and the instrument takes, on average, about two minutes to complete [22].

Globally, psychometric testing for the EQ-5D-5L with adult populations has been extensive and includes the development of UK as well as many additional normative value sets in, for example, Spain, Hong Kong, France, Ethiopia, USA, Japan, Korea, Netherlands and China [21]. A recent systematic review examined the psychometric properties of the EQ-5D-5L and reported on 99 studies across 32 countries. The systematic review reported that reliability was strong for 9 of the 9 studies that examined this property and concluded that the EQ-5D-5L had excellent psychometric properties across multiple settings, populations and conditions [23]

Specific to Australia, an EQ-5D-5L value set has been recently developed for Australian adults, and this includes establishing utility weights for each health state [24,25]. The validity of the EQ-5D-5L in Australia has been evaluated across multiple studies, and examples of these studies include testing psychometric properties of the EQ-5D-5L with the First Nations peoples of Australia [26], validation for people living with multiple sclerosis [27], validation for people undergoing a hip replacement [28] and validation for people experiencing chronic fatigue syndrome [29]. Finally, the EQ-5D-5L has also been comparted to other QOL tools within the Australian context [30].

The CHU9D instrument was developed from the ground up to measure and value QOL in child populations. The CHU9D includes nine dimensions (worried, sad, pain, tired, annoyed, schoolwork, sleep, daily routine and ability to join in activities), with five response levels attached to each dimension [14]. The CHU9D has been validated for children aged 6 to 17 years old and takes about two minutes to complete.

Globally, psychometric testing for the CHU-9D with adolescent populations has been extensive and includes the development of UK and additional normative value sets in Peru [31], Spain [32], Sweden [33] and Netherlands [34], as well as measuring the impact of specific conditions on adolescent QOL, such as inflammatory bowel disease (Canada) [35] and oral/dental health conditions (Netherlands and New Zealand) [36,37]. Validity, reliability, feasibility and overall intelligibility/acceptability have been tested in the context of Swedish school-aged children [33], Danish children receiving mental health care (longitudinal, discriminant and convergent validity) [38,39] and in the UK education setting [40], and the tool has been evaluated for appropriateness of inclusion in economic evaluations [41].

Specific to Australia, a CHU9D value set has been developed for Australian adolescents [42], and the construct validity of the CHU9D has been evaluated across multiple studies. Examples of these studies include practicality and validity with general Australian adolescents [41], testing the validity of the CHU9D as a routine outcome measure for use in child and adolescent mental health services [43], reliability, acceptability, validity and responsiveness of the CHU9D for Australian children who are overweight and obese [44], data quality, feasibility, acceptability and construct validity [45], as well as via comparison with other adolescent quality of life tools [45,46].

For both QOL instruments, the raw QOL scores were converted into utilities on the 0–1 quality-adjusted life year (QALY) scale where 0 represents a state equal to being dead and 1 represents a state of full health [47]. CHU9D utilities were generated by applying the Australian adolescent preference weighted scoring algorithm (https://licensing.sheffield.ac.uk/product/CHU9D; accessed on 1 May 2022), and this was based on the ‘profile case best worst scaling’ methods to generate adolescent population health state values [47]. The EQ-5D-5L utilities were generated by applying the UK adult preference weights via the crosswalk calculator (https://euroqol.org/eq-5d-instruments/eq-5d-5l-about/valuation-standard-value-sets/crosswalk-index-value-calculator/; accessed on 1 May 2022), and this was based on a protocol that combined the composite time trade-off valuation technique and a discrete-choice experiment [21].

### 2.6. Data Coding

Data were time-stamped and coded according to the level of government-imposed restrictions related to the location of school learning at the time of the child’s COVID-19 test. Restriction levels were categorised as Level 1: Children attend school on site (plus or minus other restrictions) or Level 2: Children do not attend school on-site and complete remote learning at home (plus or minus other restrictions).

### 2.7. Analysis

As QOL data do not typically have a normal distribution due to the positive skew towards a higher utility index score, QOLs were summarised using median and interquartile range. Median QOL between the two groups (Level 1 and Level 2) were compared using non-parametric quantile regressions to determine the effect of government-imposed restrictions, which impacted schooling location (on-site versus home), on QOL in children and parents adjusting for age, gender, symptoms of COVID-19 and chronic conditions. The QOLs were also categorised into a binary variable using the appropriate population norm as a cut-off and summarised using numbers and percentages. Binary logistic regressions were used to assess the effect of on-site schooling and home schooling on QOLs adjusting for age, gender, symptoms of COVID-19 and chronic conditions. Subsequently, the adjusted differences between two proportions were calculated using the fitted models. The child’s age, symptoms of COVID-19, gender and chronic health conditions were included in the models as these are known predictors of QOL [48,49,50].

To allow comparison with the published literature, where QOL data are most commonly presented as a mean value, the utility index and the individual domain scores have also been reported using means and standard deviations via a one-sample *t*-test for the QOL utility index scores (nominal data) as well as the individual domain scores (ordinal data). Data analysis was conducted in SPSS Version 25.

## 3. Results

There were approximately 70,000 presentations to the RIC during the study period. QOL data at the point of PCR COVID-19 testing were collected from 7931 families (*n* = 7931/70.000; 11.3%) over a time period of 71 weeks during the COVID-19 pandemic (July 2020 to November 2021). This included 2025 complete CHU9D responses for children and 5751 complete EQ-5D-5L responses for parents. Surveys missing complete CHU9D data (*n* = 74) and EQ-5D-5L data (*n* = 79) were excluded (Figure A1).

For both the CHU9D group and the EQ-5D-5L group, approximately half the children were female, and 74 per cent of children were under six years old (EQ-5D-5L group). Eighty-four per cent of children six years and over (CHU9D group) presented with COVID-19-related symptoms. The most common symptom in both groups was a runny/stuffy nose, occurring in 52 per cent under six years old and 65 per cent six years and over (Table 1).

### 3.1. CHU9D

The median CHU9D value for the on-site schooling group was 0.741 (IQR 0.549 to 0.888), and for the home schooling group, it was 0.744 (IQR 0.550 to 0.905). There was insufficient evidence to suggest a difference in median CHU9D utilities between the two cohorts after adjusting for age, gender, symptoms of COVID-19 and chronic conditions (adjusted difference in median −0.02, 95% CI −0.05 to 0.01, *p* = 0.235). The median quality of life of the on-site schooling (0.741) and home schooling (0.744) was less than the median quality of life (0.86) of the corresponding pre-pandemic population norms. The median quality of life across all age groups for both cohorts is presented in Figure 2.

During on-site schooling, 69.6 per cent (471/677) of the utilities, and during home schooling, 67.2 per cent (906/1348) of the utilities were below the 0.86 population norm score [51]. The difference in the proportion of children with a CHU9D utility below 0.86 [51] between on-site and home schooling groups was minimal after adjusting for age, gender, symptoms of COVID-19 and chronic conditions (adjusted difference in proportion 0.5%, 95% CI −3.8% to 4.9%, *p* = 0.813).

### 3.2. CHU9D Presented as Parametric Data to Enable Comparison with the Literature

At the time of COVID-19 testing, based on proxy reporting via the CHU9D by the parents of children aged 6 to 17 years old, the children’s mean utilities was 0.710 (SD 0.220). There was no mean difference in utilities between the two levels of government restrictions; however, three of the individual domains on the CHU9D had sufficient evidence of differences (Table A1). Pain was higher during on-site schooling compared to home schooling (difference in mean 0.085, 95% CI 0.018 to 0.151, *p* = 0.013), tiredness was higher in on-site schooling compared to home schooling (difference in mean 0.151, 95% CI 0.053 to 0.249, *p* = 0.003) and activity was higher in on-site schooling compared to home schooling (difference in mean 0.173, 95% CI 0.042 to 0.305, *p* = 0.010).

### 3.3. EQ-5D-5L

The median value for the on-site schooling group was 0.879 (Q1 0.837; Q3 1.000), and for the home schooling group, it was also 0.879 (Q1 0.837; Q3 1.000). There was no evidence of a difference in median EQ-5D-5L score between the two groups after adjusting for age, gender, symptoms of COVID and chronic conditions (adjusted difference in median 0, 95% CI −0.002 to 0.002, *p* = 1.000). For parents of children, the median quality of life for the current cohort (0.897) was less than the median quality of life of the corresponding pre-pandemic population norms (0.95). The median quality of life for parents of children across all child age groups is presented in Figure 3.

During on-site schooling, 59.0 per cent (1006/1705) of the utilities, and during home schooling, 60.6 per cent (2453/4046) of the utilities were below the 0.95 population norm score [25]. The difference in the proportion of parents with EQ-5D-5L utilities below 0.95 between the on-site and home schooling group was minimal after adjusting for age, gender, symptoms of COVID-19 and chronic conditions (adjusted difference in proportion 1.8%, 95% CI −0.9% to 4.6%, *p* = 0.195).

### 3.4. EQ-5D-5L Presented as Parametric Data to Enable Comparison with the Literature

At the time of the child’s PCR COVID-19 test, based on the EQ-5D-5L, the combined mean utility index for parents of children aged five years and under was 0.882 (SD 0.131). For the EQ-5D-5L, there was no mean difference in utility index score between the two levels of government restrictions; however, one of the individual domains on the EQ-5D-5L had a significant difference; Table A2. Anxiety was lower during on-site schooling compared to home schooling (difference in mean −0.065, 95% CI −0.108 to −0.021, *p* = 0.003).

## 4. Discussion

Children and their parents accessing PCR COVID-19 testing experienced a lower QOL at the point of testing when compared to pre-pandemic population norms. However, there was no observed effect of schooling location, impacted by government-imposed COVID-19 pandemic restrictions, on QOL in children or parents. These findings indicate that during COVID-19 testing, parent and child QOL utilities were found to be between 0.15 and 0.07 below normative data, exceeding the reported 0.03 minimal clinically important difference for generic QOL utility scores [52,53,54].

The difference in utility scores could be due to multiple factors, including the threat of the COVID-19 pandemic itself or the response, such as the government-imposed restrictions, changes in employment or living arrangements, COVID-19 testing process or other individual reasons. There have been many disproportionate effects of the pandemic on the wellbeing of children, which may have impacted the utility scores [6,9,50,55]. Young people have experienced an increase in mental health difficulties and reported a reduction in physical activity, healthy eating and general wellbeing [9,49,55]. An Australian study found a decrease in life satisfaction and an increase in anxiety and depression among parents and children during lockdown [56].

We do not understand to what extent COVID testing and the period of uncertainty following testing impact children’s QOL, but isolation after testing due to a positive or suspected COVID diagnosis likely results in a QOL utility decrease in parents and children [56,57]. There is minimal data regarding the impact of the COVID testing process itself or the impact of repeated testing, which many children experienced as they returned to childcare and school. It should be noted that the testing experience has changed over time, particularly since late 2021 when the government began to recommend at-home Rapid Antigen Testing instead of, or in addition to, PCR testing.

While this study did not show that home schooling compared to on-site schooling was a factor that impacted quality of life, other independent factors, such as vaccination, did impact quality of life during the COVID-19 pandemic. Globally, vaccines improved quality of life and mental health, reduced the spread of the disease, improved health outcomes for hospitalised patients and contributed to the end of lockdowns [58,59,60,61]. Specific to quality of life, when comparing partially vaccinated people and people waiting to be vaccinated to fully vaccinated people, the fully vaccinated people had a higher quality of life [58].

As the impact of the COVID-19 pandemic differed within counties and between countries, it is important that literature reports on the impact on the whole of the country, as well as on regions within a country. Survey-based studies examining the regional impacts have focussed on specific sub-populations, and recent examples of such studies include the following: the impact of COVID-19 on reduced physical activity experienced by young adult university students in Italy (*n* = 1430) [62]; the impact of COVID-19 on vaccine hesitancy from home health care service recipients in Saudi Arabia (*n* = 426) [63]; the impact of COVID-19 on developmental outcomes among kinship foster care children in the Republic of Korea (*n* = 217) [64]; the impact of COVID-19 on of perceived travel risks by travellers in Taiwan (*n* = 500) [65] and the impact of COVID-19 related lockdown on adult mental health in Italy (*n* = 1085) [66]. The current survey-based study reports on the regional impacts of a specific sub-population, specifically parent-reported child and parent quality of life during COVID-19 testing at a tertiary paediatric hospital outpatient clinic in metropolitan Melbourne, Australia (*n* = 7776). The current study is unique in that surveys were completed by parents of young children who visited the state’s largest paediatric hospital outpatient clinic for COVID-19 testing, which included the only dedicated facility for testing children under 5 years of age, in a region that experienced the most stringent pandemic lockdown conditions in Australia and globally.

### Strengths and Limitations

This is the only study we are aware of in Australia that captured QOL data for children and families during the height of the COVID-19 restrictions in 2020–2021 in the state that experienced the most COVID-19 cases and strongest restrictions. This study has several strengths, including an extended data collection period of 71 weeks which captured the changing government-imposed COVID-19 restrictions, a large sample size for both the CHU9D and the EQ-5D-5L surveys, as well as support through Victoria’s largest public paediatric hospital.

Data from this study may enable future researchers to identify the pandemic shock over a long period of time. In addition, by using validated and frequently used QOL tools within this study [67], the opportunity for comparison between current and future studies has been maximised. This study has also highlighted the value of including QOL measures in longitudinal data sets to understand change over time and change in relation to significant public health events.

Despite the new and valuable data produced by this study, it has a number of limitations that should be considered. Our study population was not representative of the whole Victorian (or Australian) population and, as such, was skewed towards younger adults/parents (due to those under the age of five being referred to The Royal Children’s Hospital RIC for testing) who were relatively homogenous demographically. The survey was only available in English, limiting participation from culturally and linguistically diverse participants. We were unable to track multiple responses from one family, so some families could have been counted more than once. While we captured QOL around the time of COVID-19 testing, it is not possible to say whether the testing experience itself or other circumstances were the primary contributing factors to the QOL reported on the day. At the time of survey completion, participants did not know their test results.

Parents of children aged birth to five years old completed the EQ-5D-5L [21] questionnaire on their own behalf, as no QOL tool has been validated for children of this age group. Parents of children aged 6 to 17 years old completed the validated proxy-reported CHU9D questionnaire on behalf of their child [38]. The reason this was completed as a proxy by the parents, compared to being completed by the children themselves, was due to early reports of the negative impact of COVID-19 on child mental health and the desire to minimise the research burden on children. However, this proxy child QOL score may be prone to bias from the parent completing the survey. While other options for determining the child QOL may have been more accurate and less prone to bias, for example, the child completing the QOL questionnaire themselves, it was not feasible in the current study. While all parents were invited to complete the QOL survey (i.e., no sample-selection bias), it was not completed by all parents, so there may be bias with parents who did complete the survey representing those experiencing a different (higher or lower) QOL when compared to those who did not complete the QOL survey (i.e., potential for self-selection bias).

The analyses could only consider factors associated with the children, e.g., age, gender and co-morbidities, not with the parents. This is a potential limitation as these parental factors could also have impacted the results. A further limitation was that the difference between the QOL reported in the current study and the QOL reported in a previous pre-pandemic study was not tested for statistical significance, as the comparison involved two different populations, i.e., while our analysis did not test for statistical significance, it did explore how the current study median values and interquartile range compares to the previous pre-pandemic study median values (Figure 1 and Figure 2).

Finally, due to the COVID-19 pandemic, there were immense time pressures to commence the COVID Wellbeing program of research. As such, time did not permit the involvement of patient groups in the design or conduct of the study. Generalisability is limited to the metropolitan population in Victoria due to the single point of data collection.

In the future, we recommend that the preservation of community quality of life is given equal importance to other key pandemic considerations, such as reducing the spread of the disease and developing vaccinations to prevent infections and improve health outcomes for hospitalised patients.

## 5. Conclusions

Children and their parents accessing PCR COVID-19 testing experienced a lower QOL at the point of testing when compared to pre-pandemic population norms. However, there was no observed effect of schooling location, impacted by government-imposed COVID-19 pandemic restrictions, on QOL in children or parents. This large-scale study assessing the QOL of school-aged children and of parents of pre-school children during the COVID-19 pandemic provided unique insights into the impact of testing and government-imposed restrictions to inform future decision-making and planning globally in relation to COVID-19 and future pandemics. This study also highlighted the importance of collecting QOL in longitudinal data sets.

## Figures and Tables

**Figure 1 healthcare-11-02555-f001:**
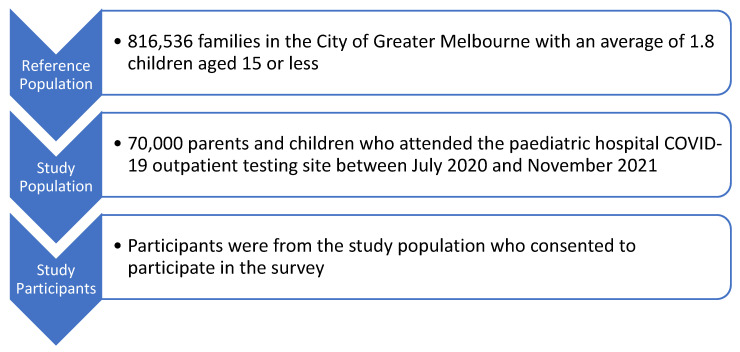
Sample selection.

**Figure 2 healthcare-11-02555-f002:**
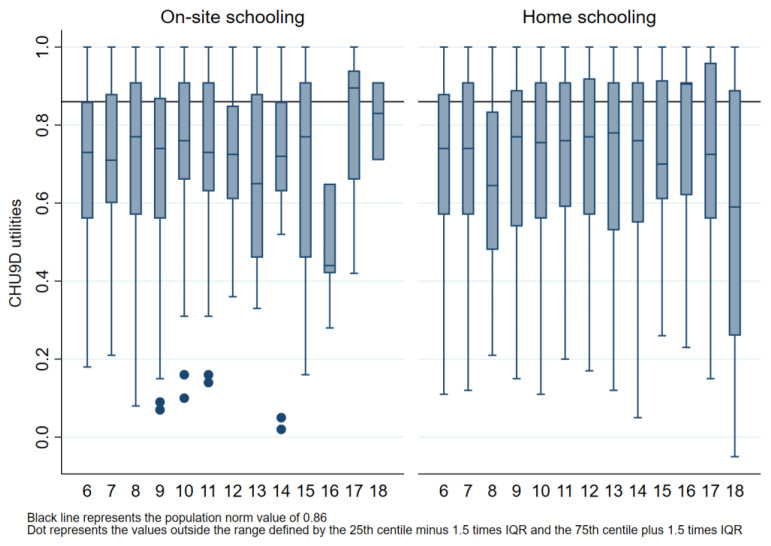
Comparison of median (IQR) CHU9D utility scores for all age groups by on-site or home schooling and by the population norm. Box plot lines represent 25th, 50th and 75th percentiles.

**Figure 3 healthcare-11-02555-f003:**
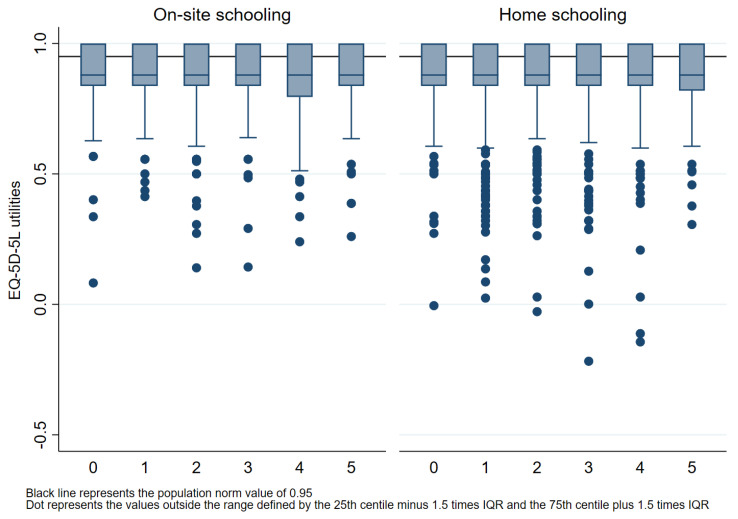
Comparison of median (IQR) EQ-5D-5L utility scores for all age groups by on-site or home schooling and by the population norm. Box plot lines represent 25th, 50th and 75th percentiles.

**Table 1 healthcare-11-02555-t001:** Demographic and baseline symptoms of children included in study.

	CHU9D (*n* = 2025)	EQ-5D-5L (*n* = 5751)
** Age, mean (SD) **	8.41 (3.63)	2.78 (1.74)
** Female, *n* (%) **	980 (48.4%)	3002 (52.2%)
** Chronic conditions Yes, *n* (%) **	241 (11.9%)	274 (4.8%)
** COVID Symptoms, *n* (%) * **		
Fever	186 (9.2%)	1120 (19.5%)
Chills or shakes	54 (2.7%)	127(2.2%)
Cough	696 (34.4%)	3070 (53.4%)
Diarrhea	21 (1.0%)	146 (2.5%)
Sore throat	789 (39.0%)	807 (14.0%)
Breathing difficulties	47(2.3%)	221 (3.8%)
Loss of appetite	70 (3.5%)	431 (7.5%)
Headache	240 (11.9%)	136 (2.4%)
Muscle ache	61 (3.0%)	38 (0.6%)
Abdominal pain	77 (3.8%)	94 (1.6%)
Stuffy or runny nose	1051 (51.9%)	3708 (64.5%)
Loss of smell	29 (1.4%)	11 (0.2%)
Nausea	79 (3.9%)	98 (1.7%)
No symptoms	531 (26.2%)	918 (16%)

* Reported by the parent at the time of PCR testing.

## Data Availability

The data presented in this study are available on reasonable request from the corresponding author. The data are not publicly available due to privacy restrictions.

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
