# Peer review of "Parent-Reported Child and Parent Quality of Life during COVID-19 Testing at an Australian Paediatric Hospital Outpatient Clinic: A Cross-Sectional Study"

_healthcare, 2023, doi:10.3390/healthcare11182555_

Round 1

Reviewer 1 Report (Previous Reviewer 1)

The authors have improved the article, and described their findings more clearly.

As a matter of personal preference, I still think there are too many figures that are not necessarily providing additional information. However, since this is a preference I will not insist that the authors agree with me.

At this point, I can recommend publication.

Author Response

Many thanks for your feedback.

Your reviewer comments have been addessed directly into the manuscript.

Warm regards, A/Prof Brusco

Reviewer 2 Report (Previous Reviewer 2)

Thank you for the opportunity to review this manuscript once again. The authors have done extensive work to address all the issues raised by the reviewers and the editor. I believe that the manuscript has significantly improved. The methods are more clear and in the context of the current reality. In addition, the results have improved and the discussion addresses the main strengths and limitations of the work.

I think that the main remaining issue is the results section. 

First, if there is no difference in CHU9D between the groups in the univariate analysis, why using a regression model? It is obvious that it would also not be significant (lines 286-288). This issue is similar for the EQ-5D-5L section.

Second, the fact that the QOL was lower in almost all age groups compared with the pre-pandemic median do not say much. You should say what the IQR is and compare it with 0.86 using a statistical test and report the p-value. It could be that the margins of all results include 0.86 and in that case none are different from it. This issue is similar for the EQ-5D-5L section.

Third, why are you dividing your results by age groups (lines 289-294). I think you should first compare the cohort's QOL to the 0.86 and only then show a sub-analysis by age groups.

Fourth, in the EQ-5D-5L section you write that they were below the 0.95 – is it statistically significant?

Fifth, in the section - "3.2. CHU9D presented as parametric data to enable comparison with the 333 literature" – please add the values and p value after you state something was higher. This issue is similar for section "3.4. EQ-5D-5L presented as parametric data to enable comparison with the literature".

No major english editing required.

Author Response

Many thanks for your feedback.

Your reviewer comments have been addessed directly into the manuscript.

Warm regards, A/Prof Brusco

Round 2

Reviewer 2 Report (Previous Reviewer 2)

I thank the authors for their hard work and addressing all the issues I raised in my previous review. I think the results section has been significantly improved. In my view, the only remaining issue is the main finding of the difference between the QOL reported by the authors to the pre-pandemic QOL. I agree that these are different populations but you still compare them to each other. I think that at the minimum the authors need to describe it as a main limitation. The whole point of using a statistical test (which addresses the variability of the results as well) is to validate this difference between the QOLs and see that this difference is not by chance. This is critical and if their analysis would have included the same cohort (with pre and post pandemic estimations) I guess the authors would do such statistical analysis on their own.  

See above.

Author Response

Dear Reviewer,

Many thanks for your ongoing support and guidance with this manuscript. Based on your advice, we have expanded the limitation section to include the following:

"A further limitation was that the difference between the QOL reported in the current study and the QOL reported in a previous pre-pandemic study was not tested for statistical significance, as the comparison involved two different populations, i.e., while our analysis did not test for statistical significance, it did explore how the current study median values and interquartile range compares to the previous pre-pandemic study median values (Figures 1 and 2)."

Warm regards, A/Prof Brusco

This manuscript is a resubmission of an earlier submission. The following is a list of the peer review reports and author responses from that submission.

Round 1

Reviewer 1 Report

This research examined parents' view of their own quality of life (for children under five) or their childrens' quality of life (6-17): parents were sampled when bringing their children in for COVID testing. The sample size is extremely large, giving the study considerable power.

1. Australian location: the authors mention (more than once) the importance of the setting of their research (Victoria, Australia). Perhaps this should be included in their title?

2. Line 138. Data were sampled relatively late in the epidemic? Also the authors refer to some factors related to Australian epidemic policy. Perhaps a paragraph locating the research in these respects should be added.

3. Line 165. Were the calculators mentioned computing a linear transformation of scores from the full range to a 0-1 range? Some added detail is needed at this point (beyond naming the calculators).

4. Line 179. Do the two types of schooling represent two continuous time zones or did regulations flip back and forth during the time of testing? This could be described in the answer to point 2 above.

5. Table 1. This tables reveals what I am quite sure are significant differences in COVID symptoms between the two groups (0-5 and 6-17). For example, the symptom of fever occurs considerably more frequently in the younger age group while the symptom of sore throat occurs more often in the older age gorup. The authors should test and discuss this.

6. Figures 2 and 4. These are confusing figures and they do not communicate much because of the difference in sample sizes. I believe a mention of the % above or below the standard of .86 is sufficient to convey the intended information. The authors might also wish to employ a z test of a proportions to compare the two proportions.

7. Figure 1. This figure mentions a Mann-Whitney test that is NOT referenced within the article. It should be. Again, a difference in sample size makes it difficult to read these tables visually. They should be changed to percentages or omitted entirely.

8. Line 226. How were the data adjusted for age, gender, etc.? The type of analysis (likely a logistic regression) is not mentioned, nor are p values provided for the conclusion.

9. Line 228 refers to a "reduction" in quality of life. These are two distinct groups, not one group changing over time. In fact one group is the test standardization group.

10. Lines 260-268. Line 292. The type of test employed to reach these conclusions is not mention, and neither are the p values.  This information should be included.

11. Figures overall: There are too many uninformative figures in the article. Unless they are discussed in a more detailed manner, all four of the figures in the main text should be removed.

12: Effect sizes are not mentioned, nor are the sizes of such differences as are present discussed (e.g. as being weak or strong).

Overall Recommendations: The information in this article is valuable for two reasons -the size of the sample and its nature. I would advise the authors to revise the article, describe their analyses more fully, and avoid using figures that convey little to the reader.

Author Response

Reviewer #1

This research examined parents' view of their own quality of life (for children under five) or their childrens' quality of life (6-17): parents were sampled when bringing their children in for COVID testing. The sample size is extremely large, giving the study considerable power.

  1. Australian location: the authors mention (more than once) the importance of the setting of their research (Victoria, Australia). Perhaps this should be included in their title?

            RESPONSE: Thankyou for this suggestion, the title now includes the word Australian, i.e., “Measuring child and parent Quality of Life during COVID-19 testing at an Australian paediatric hospital: cross sectional study”

  1. Line 138. Data were sampled relatively late in the epidemic? Also the authors refer to some factors related to Australian epidemic policy. Perhaps a paragraph locating the research in these respects should be added.

            RESPONSE: Section 2,1 and 2.4 has been expanded to include these important points.

2.1 now states: “2.1. Study setting: We completed this study at the Royal Children’s Hospital RIC in Melbourne, Victoria. The RIC was established in May 2020 and was a large testing site in Melbourne, Victoria for children, particularly those under the age of 5 years, who required testing according to state guidelines (Victorian State Government (Department of Health), 2022). In Victoria, government-imposed restrictions relating to stay-at-home orders, including home-schooling, were the most stringent in the country, and the stay-at home orders were ceased and then reenacted multiple times throughout the COVID-19 pandemic”.

2.4 now states “2.4. Data collection: Data was collected over 18 months, between July 2020 and November 2021, representing the early to mid-stages of the COVID-19 pandemic. The surveys were administered using REDCap (Harris et al., 2019; Harris et al., 2009), a secure web-based platform to capture data for research, hosted at Murdoch Children’s Research Institute, Melbourne. Parents of children aged 0-5 years completed the EuroQOL 5-Dimension 5-Level (EQ-5D-5L) (EuroQOL, 2020) questionnaire on their own behalf. Parents of children aged 6-17 years completed the validated proxy-reported Child Health Utility 9 Dimension (CHU9D) (Petersen et al., 2018) questionnaire on behalf of their child.”

  1. Line 165. Were the calculators mentioned computing a linear transformation of scores from the full range to a 0-1 range? Some added detail is needed at this point (beyond naming the calculators).

            RESPONSE: This is a non-linear transformation, as each of the domains has a different impact on the final utility weight when combined via the algorithm. To clarify this point, we have expanded this paragraph to include the following: “For both QOL instruments, the raw QOL scores were converted into utilities on the 0-1 QALY scale where 0 represents a state equal to being dead and 1 represents a state of full health (Ratcliffe et al., 2016). CHU9D utilities were generated by applying the Australian adolescent preference weighted scoring algorithm https://licensing.sheffield.ac.uk/product/CHU9D and this was based on the “profile case best worst scaling” methods to generate adolescent population health state values (Ratcliffe et al., 2016). The EQ-5D-5L utilities were generated by applying the UK adult preference weights via the crosswalk calculator https://euroqol.org/eq-5d-instruments/eq-5d-5l-about/valuation-standard-value-sets/crosswalk-index-value-calculator/ and this was based on a protocol that combined the composite time trade-off valuation technique and a discrete-choice experiment (EuroQOL, 2020).

  1. Line 179. Do the two types of schooling represent two continuous time zones or did regulations flip back and forth during the time of testing? This could be described in the answer to point 2 above.

            RESPONSE: Yes, they did flip back and forward. Per point 2 above:

2.1 now states: “2.1. Study setting: We completed this study at the Royal Children’s Hospital RIC in Melbourne, Victoria. The RIC was established in May 2020 and was a large testing site in Melbourne, Victoria for children, particularly those under the age of 5 years, who required testing according to state guidelines (Victorian State Government (Department of Health), 2022). In Victoria, government-imposed restrictions relating to stay-at-home orders, including home-schooling, were the most stringent in the country, and the stay-at home orders were ceased and then reenacted multiple times throughout the COVID-19 pandemic”.

  1. Table 1. This tables reveals what I am quite sure are significant differences in COVID symptoms between the two groups (0-5 and 6-17). For example, the symptom of fever occurs considerably more frequently in the younger age group while the symptom of sore throat occurs more often in the older age group. The authors should test and discuss this.

            RESPONSE: In table 1 we presented the characteristics to describe the study sample by age groups . Hence, we reported summary statistics i.e mean (SD) for continuous variable and number (percent) for categorical variables. These statistics are adequate to describe the differences in sample characteristics between the two cohorts. Using inferential statistics like standard error, confidence intervals, p-value based statistical tests would imply that we are trying to infer the findings about the population, which isn’t true and wasn’t the purpose of Table 1. Our presentation of Table 1 without comparing the baseline characteristics statistically is also in line with the STORBE guidelines (PLoS Med. 2007;4(10):e296. PMID: 17941714).

  1. Figures 2 and 4. These are confusing figures and they do not communicate much because of the difference in sample sizes. I believe a mention of the % above or below the standard of .86 is sufficient to convey the intended information. The authors might also wish to employ a z test of a proportions to compare the two proportions.

            RESPONSE: We thank the reviewers for their suggestions and have removed Figures 1 to 4 since the key findings are described in the text with appropriate statistics. We have now included a revised figure presenting the median QOL for all age groups and by schooling type and by indicating the standards.

            As mentioned in the methods child’s age, symptoms of COVID-19, gender, and chronic health conditions were predictors of QOL. Hence, instead of simply comparing the proportions between two groups using z-test, we used binary logistic regression and later used the fitted model to estimate the adjusted difference in proportion.

  1. Figure 1. This figure mentions a Mann-Whitney test that is NOT referenced within the article. It should be. Again, a difference in sample size makes it difficult to read these tables visually. They should be changed to percentages or omitted entirely.

            RESPONSE: We thank the reviewers for their suggestions and have removed Figure 1.

  1. Line 226. How were the data adjusted for age, gender, etc.? The type of analysis (likely a logistic regression) is not mentioned, nor are p values provided for the conclusion.

RESPONSE: We apologies for the confusion. As mentioned in the methods child’s age, symptoms of COVID-19, gender, and chronic health conditions were predictors of QOL. These variables were included in the quantile regression models for continuous variables and in the binary logistic regression models for the binary variables as predictors of the outcome. We have revised the data analysis section accordingly.

We have now reported the adjusted difference in median and proportions along with 95% confidence intervals and p-values in the results.

  1. Line 228 refers to a "reduction" in quality of life. These are two distinct groups, not one group changing over time. In fact one group is the test standardization group.

            RESPONSE: Thankyou, we appreciate this correction. The word “reduction” has been replaced with “less than”.

  1. Lines 260-268. Line 292. The type of test employed to reach these conclusions is not mention, and neither are the p values.  This information should be included.

            RESPONSE: Please refer to the response to Point 8.

  1. Figures overall: There are too many uninformative figures in the article. Unless they are discussed in a more detailed manner, all four of the figures in the main text should be removed.

            RESPONSE: As mentioned in response to the previous comment, we have now removed Figure 1 to 4  since the key findings are described in the text with appropriate statistics. We have now included a revised figure presenting the median QOL for all age groups and by schooling type and by indicating the standards.

12: Effect sizes are not mentioned, nor are the sizes of such differences as are present discussed (e.g. as being weak or strong).

RESPONSE: We apologies for this mistake. We have now reported the adjusted difference in median and proportions along with 95% confidence intervals and p-values in the results.

Overall Recommendations: The information in this article is valuable for two reasons -the size of the sample and its nature. I would advise the authors to revise the article, describe their analyses more fully, and avoid using figures that convey little to the reader.

            RESPONSE: Thankyou for this feedback.

Reviewer 2 Report

Thank you for the opportunity to review this manuscript. This is a well-performed study with large sample and it addresses a very interesting and important topic. In general, the authors describe the QUL in children age 6-17 and parents of children age 0-5 during the COVID-19 pandemic (on day of testing) and evaluate its associations with place of school. There are few issues that should be addressed in my view:

1.      Did parents fill the questionnaire while they were in the RIC? Or were they able to fill it later at home? In either case, how can you be sure they filled it on behalf of their 6-17 y.o. child?

2.      The authors should add additional analyses. First, a general analysis of predictors for lower QOL scores in each of the questionnaires (do age, gender, symptoms during filling the questionnaire and others are associated with lower QOL?). Second, is there a specific age that home vs. on-site schooling was associated with a change in reported QOL?

3.      I am not sure what is the added value of presenting figures 1, 2, and 4. The first is a statistical analysis that is not usually presented and the other two do not add much data given the high amount of patients (dots in the figures) that are presented.

4.      Did you analyzed how many of the participants (children or parents) had prior COVID-19? How many were vaccinated? It could possibly have an effect on the QOL.

5.      I think the authors should include a paragraph addressing ways to improve QOL in their study population. In my opinion vaccines were a crucial aspect as they limited the spread of the disease, improved outcomes and enabled an earlier time until the lock-down was canceled. I recommend the authors to use the following paper in their discussion which describe the mentioned above: https://pubmed.ncbi.nlm.nih.gov/35536849/

Good level of English

Author Response

Reviewer #2

Thank you for the opportunity to review this manuscript. This is a well-performed study with large sample and it addresses a very interesting and important topic. In general, the authors describe the QUL in children age 6-17 and parents of children age 0-5 during the COVID-19 pandemic (on day of testing) and evaluate its associations with place of school. There are few issues that should be addressed in my view.

  1. Did parents fill the questionnaire while they were in the RIC? Or were they able to fill it later at home? In either case, how can you be sure they filled it on behalf of their 6-17 y.o. child?

            RESPONSE: In general parents filled this in at the RIC. The methods has been updated to state “Most questionnaires were completed at the RIC during the visit, as compared to being completed at home after the RIC visit”.

In addition, the CHU9D explicitly refers to the health state of the child in each question, hence we are confident that it refers to the child, not the parents. 

  1. The authors should add additional analyses. First, a general analysis of predictors for lower QOL scores in each of the questionnaires (do age, gender, symptoms during filling the questionnaire and others are associated with lower QOL?). Second, is there a specific age that home vs. on-site schooling was associated with a change in reported QOL?

            RESPONSE: We appreciate the reviewers’ suggestions. The aim of this study was to investigate the effect of home schooling on QOL compared to on-site schooling during the pandemic not identifying the predictors of the lower QOL which has already been reported in previous literature (Chen et al., 2020; Fuerboeter et al., 2021; Riiser et al., 2020).

            We have now included a revised figure presenting the median QOL for all age groups and by schooling type and indicated the standards to explore the association between age and median QOLs.

  1. 3.      I am not sure what is the added value of presenting figures 1, 2, and 4. The first is a statistical analysis that is not usually presented and the other two do not add much data given the high amount of patients (dots in the figures) that are presented.

            RESPONSE: As mentioned in response to previous comments, we have removed Figures 1 to 4.

  1. Did you analyzed how many of the participants (children or parents) had prior COVID-19? How many were vaccinated? It could possibly have an effect on the QOL.

            RESPONSE: This is a valid question, but unfortunately we do not have this data.

  1. I think the authors should include a paragraph addressing ways to improve QOL in their study population. In my opinion vaccines were a crucial aspect as they limited the spread of the disease, improved outcomes and enabled an earlier time until the lock-down was canceled. I recommend the authors to use the following paper in their discussion which describe the mentioned above: https://pubmed.ncbi.nlm.nih.gov/35536849/.

            RESPONSE: This is an insightful reflection, thankyou. The following has been added to the discussion section.

“This study did not show a significant difference between home schooling, compared to on-site schooling, however quality of life scores were already significantly lower than pre-pandemic norms. This is in contrast to previous studies that did find home schooling impacted quality of life during the pandemic [2, 16]. Viner and colleagues completed a systemic review, and this included papers across 20 countries, and found that school closures had a negative impact on the health and wellbeing of children and young people [2].

In addition, there are other independent factors, such as vaccination, that impacted quality of life during the COVID-19 pandemic [35, 41]. Globally, vaccines improved quality of life and mental health, reduced the spread of the disease, improved health outcomes for hospitalized patients and contributed to the end of lock-downs [35, 41-43]. Specific to quality of life, when comparing partially vaccinated people and people waiting to be vaccinated, to fully vaccinated people, the fully vaccinated people had a higher quality of life [35].”

Reviewer 3 Report

This manuscript presents a study on Quality of Life during COVID-19 in the paediatric setting. This is an interesting, important, and recent topic, although it is already losing its timeliness. The study has some positive aspects, such as having two large samples (N=2,025; N=5,751) and balanced by gender.

However, some aspects could be further explored, better explained, and linked throughout the manuscript. A critical appraisal of each of these sections and suggestions for improvement are presented below.

1. Title

The title refers to the paediatric setting but does not specify who the study sample is. Whose quality of life (paediatricians, parents, children...)?

2.  Abstract

The abstract summarizes each of the sections of the study.

Just a short comment. The authors start by mentioning that the impact of the pandemic on the quality of life of adults is known, but they do not refer to the quality of life of children.

The idea they refer to is equally valid for the paediatric context and this is the specificity of this study.

3. Introduction

The introduction recalls the constraints imposed by the pandemic and its consequences on the quality of life of adults. It highlights the importance of knowing this phenomenon in the Australian state of Victoria.

However, the introduction requires further problematisation about the need to know the impact on children's quality of life, as well as why to investigate the effect of schooling location (on-site or home-based).

4. Method

The method is described in a detailed and rigorous manner.

The authors explore only part of the results of a larger study. Since they have more data available, it would have been interesting to be able to explore the relationship of quality of life with other variables.

Throughout the reading of the manuscript, there is some repetition of information. There is a repetition of information, regarding the age range of the children, the completion of each instrument, the time span of application. This information is presented at the end of the Introduction, in the Sample, Data collection, then in the Results,...

I recommend that the text be revised to avoid redundancy.

5.  Results

The analysis of the results should be more connected to the introduction of the study, i.e., the data is explored without a clear justification in the introduction.

6.  Discussion

I think it would be important to clarify that the results reflect parents' perceptions of their children's quality of life, which may lead to some bias. I also think that the children's quality of life could have been assessed by a combination of other methods and not only by a self-administered questionnaire.

7.  Conclusion

In the conclusion, the authors state that this study provided unique insights into the impact of testing and government-imposed restrictions to inform future decision making and planning globally, in relation to COVID-19 and future pandemics. However, this information is broad. What are the specific contributions of this study? Based on these results, what do you recommend for the future?

Author Response

Reviewer #3

This manuscript presents a study on Quality of Life during COVID-19 in the paediatric setting. This is an interesting, important, and recent topic, although it is already losing its timeliness. The study has some positive aspects, such as having two large samples (N=2,025; N=5,751) and balanced by gender. However, some aspects could be further explored, better explained, and linked throughout the manuscript. A critical appraisal of each of these sections and suggestions for improvement are presented below.

  1. Title

The title refers to the paediatric setting but does not specify who the study sample is. Whose quality of life (paediatricians, parents, children...)?

RESPONSE: Thankyou for this suggestion, the title now states “Measuring child and parent Quality of Life during COVID-19 testing at an Australian paediatric hospital: cross sectional study”

  1. Abstract

The abstract summarizes each of the sections of the study.

Just a short comment. The authors start by mentioning that the impact of the pandemic on the quality of life of adults is known, but they do not refer to the quality of life of children.

The idea they refer to is equally valid for the paediatric context and this is the specificity of this study.

            RESPONSE: Thanks for this feedback. The abstract has been updated to state:

ABSTRACT: Background: Globally we have seen a drop in adult and child quality of life (QOL) during the COVID-19 pandemic, however little is known about adult or child QOL during the height of the pandemic in the Australian context, and the impact of government-imposed restrictions, specifically on-site versus home-schooling. Aims: Our study aimed to establish if QOL in children and parents, presenting to a hospital-based Respiratory Infection Clinic in Victoria, Australia for COVID-19 PCR testing, differed from pre-pandemic population norms, and if on-site versus home-schooling further impacted QOL. Methods: Following the child’s test and prior to receiving results, consenting parents of children aged 6-17 completed the CHU9D instrument on their child’s behalf, and parents of children aged 0-5 completed the EQ-5D-5L instrument on their own behalf (cross-sectional study). Data analyses utilised quantile regression, adjusting for child’s age, COVID-19 symptoms, gender, and chronic health-conditions. Results: July-2020 to November-2021, 2,025 parents completed CHU9D; child’s mean age 8.41-years (±3.63SD) and 48.4% (n=980/2025) female; 5,751 parents completed EQ-5D-5L; child’s mean age 2.78-years (±1.74SD) and 52.2% (n=3,002/5751) female. QOL utilities were lower than pre-pandemic norms for 68% of CHU9D and 60% EQ-5D-5L scores. Comparing periods of on-site to home-schooling, there was no difference between the median CHU9D (0.017, 95%CI-0.05 to 0.01) or EQ-5D-5L scores (0.000, 95%CI-0.002 to 0.002). Conclusion: Our large-scale study found that while QOL was reduced at the point of COVID-19 testing during the pandemic, differing levels of government-imposed restrictions didn’t further impact QOL. These unique insights will inform decision-making, in relation to COVID-19 and future pandemics.

  1. Introduction

The introduction recalls the constraints imposed by the pandemic and its consequences on the quality of life of adults. It highlights the importance of knowing this phenomenon in the Australian state of Victoria.

However, the introduction requires further problematisation about the need to know the impact on children's quality of life, as well as why to investigate the effect of schooling location (on-site or home-based).

            RESPONSE: The introduction has been updated to include current references reporting on the impact of COVID-19 on children, to complement those references already there describing the impact on adults.

In addition, the following has been added to the introduction “While children and families in Victoria have been significantly affected by some of the most stringent COVID-19 related restrictions and long periods of remote school learning globally, little is known about child and parent QOL during the height of the pandemic in the Australian context, nor about the impact of home-schooling during the stay-at-home orders. It is important to investigate the impact on children's quality of life because the health state of a child can impact the QOL of the parents. It is also important to investigate the effect of home-schooling because this was a core strategy for the Victorian stay-at-home orders and placed much stress on the family unit during the COVID-19 pandemic.

  1. Method

The method is described in a detailed and rigorous manner.

The authors explore only part of the results of a larger study. Since they have more data available, it would have been interesting to be able to explore the relationship of quality of life with other variables.

Throughout the reading of the manuscript, there is some repetition of information. There is a repetition of information, regarding the age range of the children, the completion of each instrument, the time span of application. This information is presented at the end of the Introduction, in the Sample, Data collection, then in the Results,...

I recommend that the text be revised to avoid redundancy.

            RESPONSE: Thankyou, the repetition has been removed.

  1. Results

The analysis of the results should be more connected to the introduction of the study, i.e., the data is explored without a clear justification in the introduction.

            RESPONSE: Consistent with Point 3 above, the introduction has been updated to better reflect the analysis.

  1. Discussion

I think it would be important to clarify that the results reflect parents' perceptions of their children's quality of life, which may lead to some bias. I also think that the children's quality of life could have been assessed by a combination of other methods and not only by a self-administered questionnaire.

            RESPONSE: This has been reflected in the limitations section.

            “Parents of children aged 0-5 years completed the EQ-5D-5L(EuroQOL, 2020) questionnaire on their own behalf, as no QOL tool has been validated for children of this age group. Parents of children aged 6-17 years completed the validated proxy-reported CHU9D(Petersen et al., 2018) questionnaire on behalf of their child. The reason this was completed as a proxy by the parents, compared to being completed by the children themselves, was due to early reports of the negative impact of COVID-19 on child mental health and the desire to minimise the burden on children. However, this proxy child QOL score may be prone to bias from the parent completing the survey. While other options for determining the child QOL may have been more accurate and less prone to bias, for example, the child completing the QOL questionnaire themselves, it was not feasible in the current study. As the QOL survey was not completed by all parents, there may be bias with parents who did complete the survey representing those experiencing a different (higher or lower) QOL when compared to those who did not complete the QOL survey.”

  1. Conclusion

In the conclusion, the authors state that this study provided unique insights into the impact of testing and government-imposed restrictions to inform future decision making and planning globally, in relation to COVID-19 and future pandemics. However, this information is broad. What are the specific contributions of this study? Based on these results, what do you recommend for the future?

            RESPONSE: This study’s unique contribution was that (in red) “Children and their parents accessing PCR COVID-19 testing experienced a lower QOL at the point of testing when compared to pre-pandemic population norms. There was no observed effect of schooling location, impacted by government-imposed COVID-19 pandemic restrictions, on QOL in children or parents.” The null effect of schooling location on QOL was unexpected given the severity of the lock-downs in Victoria. The specific recommendation from this study has also been clarified as “In the future, we recommend that the preservation of community quality of life is given equal importance to other key pandemic considerations, such as reducing the spread of the disease and developing vaccinations to prevent infections and improve health outcomes for hospitalised patients.”